# Prevalence of Incidental Maxillary Sinus Anomalies on CBCT Scans: A Radiographic Study

**DOI:** 10.3390/diagnostics13182918

**Published:** 2023-09-12

**Authors:** Junaid Ahmed, Aditya Gupta, Nandita Shenoy, Nanditha Sujir, Archana Muralidharan

**Affiliations:** Department of Oral Medicine and Radiology, Manipal College of Dental Sciences Mangalore, Manipal Academy of Higher Education, Manipal 575001, Karnataka, India; junaid.ahmed@manipal.edu (J.A.); adityagupta@gmail.com (A.G.); nanditha.sujir@manipal.edu (N.S.); m.archana@manipal.edu (A.M.)

**Keywords:** cone beam computed tomography, maxillary sinus, pathology, polyps, well being

## Abstract

CBCT significantly impacts dental procedures and has brought significant reforms to our approach to diagnosis and treatment planning despite its limitations in differentiating soft tissues. It is an excellent imaging modality and quickly identifies sinus opacification and provides valuable insight into paranasal sinus pathologies, with considerably lower radiation exposure. The present study aimed to investigate the occurrence of maxillary sinus abnormalities in CBCT scans, identify the frequency, type, and location of these findings, and find the correlation between the distance of periapical lesions and radiographic changes in the maxillary sinus. Two examiners independently evaluated 117 patients to diagnose and classify the cases into different abnormality subtypes. The periapical lesions most closely related to the sinus were recorded. The diameters of the left and right maxillary sinus ostium and the distance of the ostium’s lower border to the sinus’s osseous floor were recorded. The findings were correlated with the age and gender of these patients. The present study reveals that sixty-one patients were diagnosed with mucosal thickening (52.1%). The sinus wall most affected by mucosal thickening was the maxillary sinus floor, followed by the medial and lateral walls. Of 19 patients with periapical lesions, 15 had maxillary sinus mucosal thickening, which is statistically significant (*p* = 0.004). The high occurrence of abnormalities in the maxillary sinus emphasizes the importance for the radiologist to comprehensively interpret the whole volume acquired in CBCT images, including the entire sinus. Incidental findings may be considered in the individual clinical context of signs and symptoms, reducing the risk of overestimating the real impact of radiographic findings.

## 1. Introduction

Incidental findings in the maxillary sinus are unexpected or unintended discoveries that are not directly related to the primary reason for conducting a cone beam computed tomography (CBCT) scan for dental imaging [1,2]. CBCT scans are often used in dentistry to visualize the oral and maxillofacial structures, including the teeth, jawbones, and surrounding tissues. However, due to the detailed nature of CBCT imaging, it is not uncommon to identify other anatomical structures or conditions [3,4,5]. Here are some examples of incidental findings that might be identified in the maxillary sinus during a CBCT scan taken for dental imaging:

Sinusitis: Inflammation or infection of the maxillary sinus can be observed on a CBCT scan, showing changes in the sinus lining and the presence of fluid or air–fluid levels [6].

Retention Cysts: These are fluid-filled sacs that can develop within the sinus due to blocked ducts. They are usually benign and may not require immediate treatment.

Polyps: Nasal or sinus polyps, which are non-cancerous growths, can sometimes be identified incidentally. They may cause congestion or sinus issues [7].

Mucous Retention Pseudocysts: These benign cystic lesions occur when the mucus becomes trapped in a blocked sinus cavity, causing the sinus lining to expand.

Anatomical Variations: Some people may have anatomical variations in the maxillary sinus, such as septa (dividing walls) or accessory ostia (additional openings). These variations are usually harmless [8].

Tumors or Masses: Rarely, tumors or masses might be identified incidentally within the maxillary sinus. Further evaluation is needed to determine such a finding’s nature and potential implications.

Foreign Bodies: Foreign objects that have entered the maxillary sinus, such as dental materials or fragments, might be detected during imaging [9].

Bone Pathologies: Conditions like fibrous dysplasia, osteomas, or cysts in the maxillary sinus area can be identified incidentally [10].

Dental Pathologies: Infections originating from dental roots can sometimes spread to the maxillary sinus, leading to sinus infections or abscesses. As a result, conditions affecting the maxillary sinuses can imitate dental problems and vice versa [11]. Radiographs are frequently used to diagnose diseases of the maxillary sinus, ranging from conventional 2D imaging, like panoramic radiograph and posteroanterior views of sinuses, to the more advanced 3D imaging modalities, like cone beam computed tomography (CBCT), computed tomography (CT), and magnetic resonance imaging (MRI). The multiplanar images acquired by CBCT allow radiologists to inspect the entire volume of the acquired image and anatomic variations or abnormalities found in the image volume [12]. The maxillary sinus area is usually within the imaging field when CBCT is indicated for dental implant site assessment or periapical pathologies [13]. Hence, maxillofacial radiologists frequently view the incidental findings in the maxillary sinus area and can assess the prevalence, which can be incidental, benign, or detrimental to the treatment plan.

It is important to note that not all incidental findings require immediate intervention. The clinical significance of each finding depends on factors such as the patient’s symptoms, medical history, and the nature of the finding itself. If an incidental finding is identified, the dental or medical professional may recommend further evaluation or referral to an appropriate specialist, such as an ear, nose, and throat (ENT) surgeon.

In a study by Cha et al. [14], using CBCT examinations, the abnormalities found were signs of sinusitis (7.5%), retention cysts (3.5%), and polypoid mucosal thickening (2.3%). In another study, the prevalence of flat mucosal thickening ranged from 23.7% to 38.1%, that of polypoid mucosal thickening ranged from 6.5% to19.4%, signs of acute sinusitis made up 3.6%, and partial and total opacification made up 12% and 7%, respectively [1].

The inability of a general dentist to detect incidental abnormalities in the maxillary sinus despite being visible in the CBCT volume is often due to their limited experience in interpreting volumetric images and their failure to conduct a thorough visual examination of the entire image [15]. The European Academy of DentoMaxilloFacial Radiology (EADMFR) and the American Academy of Oral and Maxillofacial Radiology (AAOMR) outline that an oral and maxillofacial radiologist should review and interpret the entire volume regardless of the region of interest [16].

The present study was undertaken to record the prevalence of incidental maxillary sinus abnormalities in CBCT scans, recognize the frequency, type, and location of these findings, and correlate the distance between the periapical lesions and the presence of any inflammatory changes in the maxillary sinus.

## 2. Materials and Methods

The retrospective study was conducted in the Department of Oral Medicine and Radiology, Manipal College of Dental Sciences, Mangalore. After receiving approval from the Institutional Ethical Committee, MAHE/MCODSMLR/198/2021, a statistician was consulted. Considering the power of the study to be 80%, with an alpha of 5%, the corresponding z value is Z = −1.95996398454005. Using the formula n = (*Zα*/2*d*)2 p(1 − p) with the minimum percentage difference deemed clinically significant at 8.1%, the sample size required was 117 for the study. It was advised to include 117 patient scans. 

Detailed information regarding age, gender, facial asymmetry, trauma, and history of upper respiratory infections was obtained from the patients’ past medical records. Large FOV CBCT scans were included in the study, extended from the roof of the orbits inferiorly to at least the second cervical vertebrae of the patients above 18. Scans of edentulous patients, patients referred exclusively for maxillary sinus abnormalities, young patients with incomplete sinus development, low-resolution images, and those with metallic artifacts that impaired sinus visualization were excluded. 

The CBCT images were obtained using standard exposure parameters and patient positioning protocols (field of view= 17 × 10 cm, resolution = 0.16 mm, 80–84 kvp, 10–12 mA) with a CBCT unit (Planmeca promax 3D Mid, Helsinki, Finland). Romexis version 4.6.2 software (Planmeca, Helsinki, Finland) was used. The exposure parameters were standardized for all the patients. The CBCT scans had panoramic reconstruction and multiplanar reformation views, i.e., axial, sagittal, and coronal planes. Two examiners who have completed their master’s degrees in oral radiology assessed all images to diagnose and classify them into different abnormality subtypes. In the case of a discrepancy between two observers, a third examiner intervened, and a consensus on the radiologic impression was obtained. The presence or absence of sinus abnormalities was identified using a yes/no scale. The radiology reports followed a consistent format and contained a listing of all radiographic findings used to tabulate the data in this study using Microsoft Excel.

Increased or decreased dimension of the sinus;Radiographic density changes in the cortical bone of the sinus;Partial or complete opacification of the sinus cavity;Increased thickening of the mucosa greater than 3 mm;Congenital changes (aplasia and hypoplasia);Inflammatory lesions like polyps;Fracture lines.

The patients were divided into the following age groups: (1) <20; (2) 21–30; (3) 31–40; (4) 41–50; (5) 51–60; (6) 61 years and above. 

The locations of the diagnosed abnormalities were recorded as affecting either the anterior, posterior, inferior, superior, lateral, or medial walls of the sinus based on the method by Nishimura and Iizuka [17]. The presence of periapical lesions in the upper posterior teeth was recorded. The lesions’ proximity to the sinus wall floor was classified using the method followed by Oberli et al. [18]. They were classified as Class I (near the sinus floor), Class II (in contact with the sinus floor), or Class III (overlapping with the sinus floor). A periapical lesion was recorded when the lamina dura was imperceptible or had an irregular appearance and when there was a radiolucency indicating bone destruction around the root apex. The connection between the teeth affected by periapical infection and the sinus floor was investigated through sagittal and coronal reconstructions. The findings were categorized as follows:
No contact between the root apex and the maxillary sinus existed.It appeared that one of the tooth roots was either touching the sinus floor or extending into the maxillary sinus. One of the tooth roots was either touching the sinus floor or extending into the maxillary sinus.More than two teeth appeared to be in contact with the sinus floor or protruded into the maxillary sinus.

Based on the above criteria, teeth with periapical lesions were graded based on the anatomic relationship as 1, 2, or 3, as described earlier. Because not all roots have the same spatial connection with the maxillary sinus, the root and/or the periapical lesion nearest the sinus floor was continuously recorded. Only the lesion most closely related to the sinus was recorded in cases of multiple periapical lesions near the sinuses. The above findings were all correlated with the age and gender of the patient. 

### Statistical Analysis

A Chi-square test was used to compare the patterns with gender and age using SPSS version 17.0 (SPSS Inc., Chicago, IL, USA) for data analysis. The results are considered significant at a *p*-value of <0.05.

## 3. Results

A total of 55 patients (47%) were male, and 62 (53%) were female. The most common indication for CBCT scans was dental implant site assessment, surgical planning, orthodontic scans, TMJ dysfunction, suspected oral malignancies, and trauma.

A total of 68 patients had pathologic findings in one or both sinuses, such as mucosal thickening, odontogenic cysts, fracture, hypoplasia, and bone lesions (58.1%) (Figure 1, Figure 2, Figure 3 and Figure 4). Of the 55 male patients, 33 had pathologies, like oroantral communication and inflammatory cysts in either sinus, whereas only 35 of the 62 scans of the female patients revealed radiographically abnormal findings in either sinus. The difference between the male and female patients was insignificant (Table 1).

Patients in their sixth decade of life showed a slightly higher prevalence of pathologic findings in the maxillary sinus, which was statistically significant (*p* = 0.017) (Table 2). 

A total of 61 patients were diagnosed with mucosal thickening (52.1%). The sinus wall most affected by mucosal thickening was the maxillary sinus floor, followed by the medial and lateral walls of the maxillary sinus. Of the 19 patients with odontogenic conditions causing periapical lesions, 15 had maxillary sinus mucosal thickening, and the relation was statistically significant (*p* = 0.004). The odontogenic cyst was diagnosed in five patients (4.3%), while a sinus wall fracture was recorded in two patients (1.7%). Maxillary sinus hypoplasia was diagnosed in three patients (2.6%). We did not observe any significant difference between the periapical grade and the occurrence of maxillary sinusitis, even though we had five cases of Grades 2 and 3, which meant that in five cases, there was a periapical region close to the maxillary sinus, which could induce a change in the latter. The frequency of the pathologies is represented in Figure 5. 

## 4. Discussion

CBCT, a 3D imaging modality, could be of clinical value in screening and planning paranasal surgery. Studies have reported the valuable application of CBCT for intraoperative imaging of the paranasal sinus [19]. One of the benefits of applying CBCT for imaging of the paranasal sinuses could be the lower dose compared to CT imaging. Moreover, CBCT delivers an isotropic volume resolution, facilitating the diagnosis of delicate structures in multiplanar reconstructions [20]. This study assessed the prevalence of incidental findings in the maxillary sinus using cone beam computerized tomography scans of patients referred to the department. The results show a high prevalence of abnormalities in the maxillary sinus, highlighting the need for thoroughly examining the entire volume of the CBCT image, which includes the maxillary sinus and its associated regions.

Many studies have acknowledged that interpreting CBCT images necessitates familiarity with the anatomy of the area being investigated, comprehension of the spatial relationships within the image volume, a deep understanding of the potential diseases, anatomical variations, and abnormalities that impact the maxillofacial region, and the ability to formulate a differential diagnosis with expertise [21].

For dental implant site assessment in the maxilla, the configuration and status of the maxillary sinus are essential to assess the available amount of bone. If a sinus lift is indicated, the visualization is helpful because the success rate of sinus lift procedures is crucially dependent on the configuration and status of the maxillary sinus [22]. Any disease arising from dental or dentoalveolar structures could damage the floor of the maxillary sinus, leading to sinusitis, known as odontogenic maxillary sinusitis (OMS). The maxillary sinus and the associated teeth are better revealed in CBCT than in periapical radiography [23,24]. 

Several studies have reported significant variability in the prevalence of incidental findings in the maxillary sinuses of asymptomatic subjects in multiplanar images [25,26,27]. CT is considered the gold standard for adequate maxillary sinus imaging because of its high resolution and ability to discern bone and soft tissue and detect sinus inflammation in approximately 30% of cases [28]. A CBCT study reported a prevalence of 24.6% to 56.3% [29], whereas our research saw incidental abnormalities in 68.3%. Discrepancies in these variations may be due to several factors, such as dissimilarities in the sampling criteria, variations in image interpretation, and the diagnostic criteria used. 

Our study covered a wide range of age distribution; it did not include patients under 12 years since the formation of their maxillary sinus is still incomplete, and specific abnormalities such as mucosal thickening and opacification are common findings in early childhood and not indicative of sinus disease. The most detected abnormality was mucosal thickening, observed in 52.1% of cases, especially among individuals under 20 and between 20 and 30 years old. A study carried out by Rege [30] also had a similar finding, where mucosal thickening was associated with irritation, such as odontogenic pathology or allergic phenomena. Nonvital posterior maxillary teeth, periodontal abscesses, retained roots, embedded, or impacted teeth, extensively carious teeth, and oroantral fistulae could be etiological factors in pathologies of odontogenic origin causing mucosal thickening [16]. 

Correlation between periapical lesions and maxillary sinusitis: There is a potential correlation between periapical lesions and maxillary sinusitis due to their proximity to the upper jaw [31]. The roots of the upper back teeth (molars and premolars) are situated close to the maxillary sinus floor. An advanced periapical infection in these teeth can erode the bone, separating the root tip from the sinus, and creating a pathway for disease to spread from the tooth’s root into the sinus cavity. This can lead to sinusitis and dental symptoms [32]. An infection from a periapical lesion can sometimes extend into the maxillary sinus, causing sinusitis symptoms. Conversely, maxillary sinusitis can sometimes cause referred pain to the upper back teeth, making it difficult to pinpoint the exact source of the discomfort. Not all cases of periapical lesions lead to maxillary sinusitis, and not all cases of maxillary sinusitis are caused by dental infections [33]. Proper diagnosis and treatment by dental and medical professionals are crucial to effectively differentiate and manage these conditions.

Research has demonstrated that periapical lesions close to or linked with the sinus floor can heighten the possibility of sinus infections. Moreover, more significant apical lesions can lead to an increase in the maxillary sinus and mucosal thickness. However, our study did not reveal any incidental discovery despite encountering five cases where the mucosal floor was near the root apices [34,35]. 

A study by Nurbakhsh et al. [36] found that the sinus wall thickening increased as the distance between the tooth apex or lesion and the maxillary sinuses decreased. Similarly, it was reported by Shanbhag et al. and Goller-Bulut et al. The current study did not document such findings nor find any correlation between apical proximity to the maxillary sinus and the age of the study subjects [37,38]. 

As a part of the multidisciplinary approach, it is recommended that a dentist and an ENT surgeon work as a team to provide an accurate diagnosis and appropriate treatment on time. The proximity between the maxillary molar roots associated with periapical lesions and the sinus floor could influence maxillary sinus irritation [27]. When periapical lesions were present, the anatomic relationship was ranked between 1 and 3, as described earlier. Because not all roots have the same spatial connection with the maxillary sinus, the heart or the periapical lesion nearest the sinus floor was continuously recorded.

Our study demonstrates a significant relationship between periapical lesions and sinus pathology (*p* = 0.04). The lower sinus wall was the most affected location within the sinus, which suggests possible odontogenic involvement [27]. The results of our study correlate with the study by Hauman et al. [27], who demonstrated a positive correlation between periapical lesions and sinus opacification.

Odontogenic cysts were the second most frequently found inflammatory abnormality (4.3%). This result is comparable to the findings of Bosio et al. [39] and Rhodus NL [29]. These studies examined general dental patients using plain panoramic radiography and reported a prevalence ranging from 1.4% to 9.6%. 

The third most frequent finding was sinus opacification, observed in six patients. The deviated nasal septum was observed in 6.3% of patients; this finding is consistent with Smith et al. [40], who found that 19.4% of their patients had a deviated septum, and 50.0% had mucosal thickening and sinus opacification, which is consistent with maxillary sinusitis. 

Considering the clinical implications of sinus inflammation when installing dental implants, particularly in patients with nasal septum deviation, is essential [23,24]. It is crucial to carefully assess the presence or absence of opacifications in the maxillary sinus. Opacification, which refers to clouding the sinuses, was not a finding in our study. However, it is essential to note that opacification can also result from other abnormalities such as mechanical trauma, barotraumas, and hemorrhage. While inflammation is the most likely cause, it is essential to rule out other possibilities, such as fungal sinusitis or neoplastic disease.

Raghav M et al. [41] reported that opacification could even be found in asymptomatic patients as an incidental finding in 67% of cases. Chen et al. [42] found that in cases of unilateral opacification, there was a 5.1% incidence of malignancy, a 10.4% incidence of benign tumors, and a 29.3% incidence of fungal disease. In a separate study, Kaplan et al. [43] found that patients who underwent endoscopic sinus surgery had a high incidence of mucoceles and nasal polyposis in cases of complete unilateral opacification. Additionally, in a review of 1118 CT scans of the maxillary sinus, Ahsan et al. [44] found that 28 of these had complete opacification, with 12 of these patients being further diagnosed with neoplastic disease.

One limitation of this study is the limited sample size and the need for prospective studies. We also recommend a multidisciplinary approach toward the management of such incidental findings. The present study stresses the importance of radiologists being alert and observing alterations that may not be related to the reasons for the initial examination. It is also important to note that incidental findings on radiographs should be assessed clinically as a part of proactive patient care.

## 5. Conclusions

The high occurrence of incidental findings in the maxillary sinus emphasizes the importance for dentomaxillofacial radiologists to comprehensively interpret the whole volume acquired in CBCT images, including the entire sinus, while analyzing the images of routine patients. Two-dimensional imaging techniques suffer from the superimposition of anatomic structures, making the documentation of incidental findings in the maxillary sinus difficult. Incidental findings may be considered in the individual clinical context of signs and symptoms, reducing the risk of overestimating the real impact of radiographic findings.

## 6. Patents

No patents have been filed from this study.

## Figures and Tables

**Figure 1 diagnostics-13-02918-f001:**
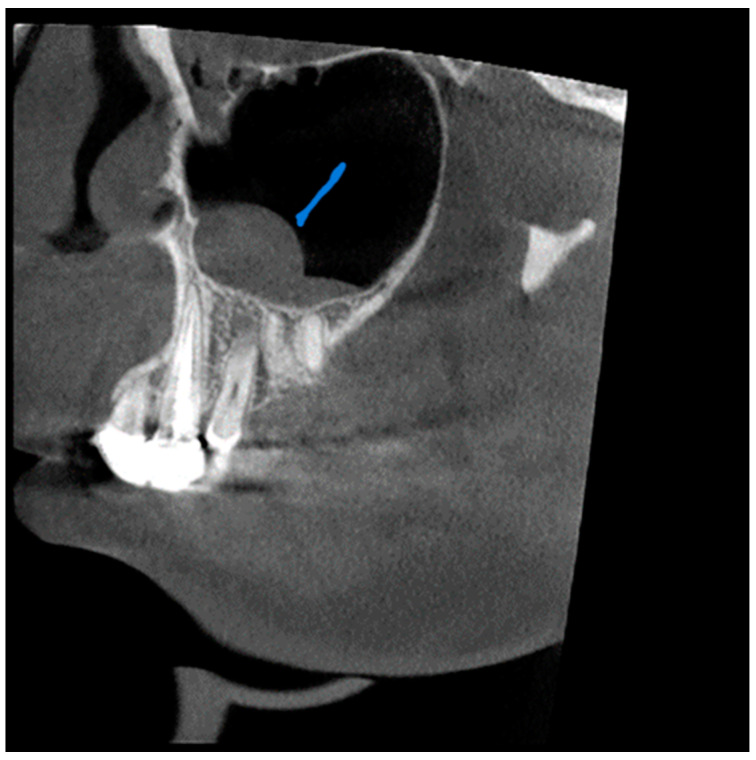
View from the sagittal plane. The Blue arrow shows shows a thickening of the mucous boundaries of the maxillary sinuses.

**Figure 2 diagnostics-13-02918-f002:**
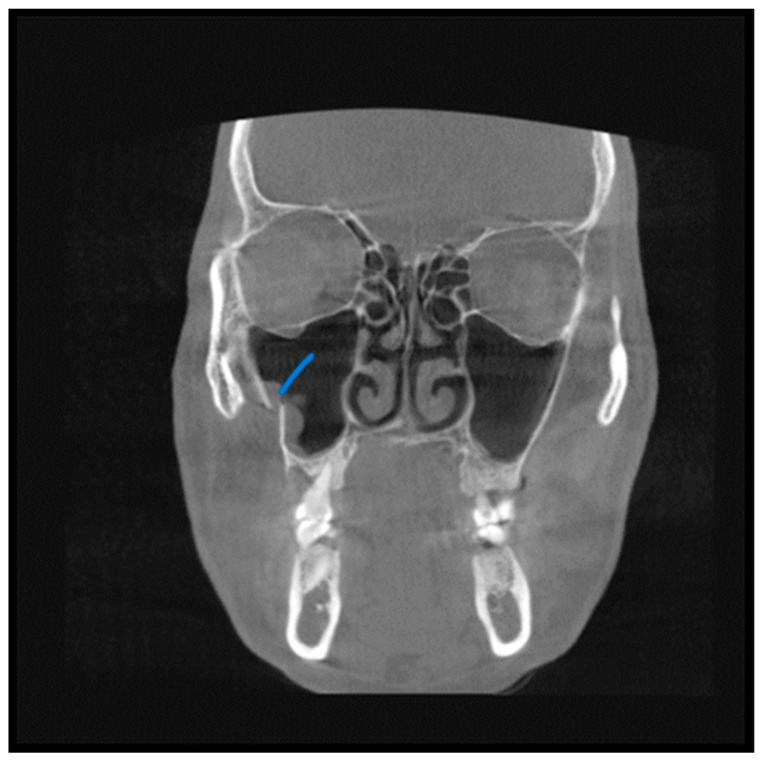
View from the coronal plane. The Blue arrow shows a fracture line along the lateral wall of the maxillary sinuses.

**Figure 3 diagnostics-13-02918-f003:**
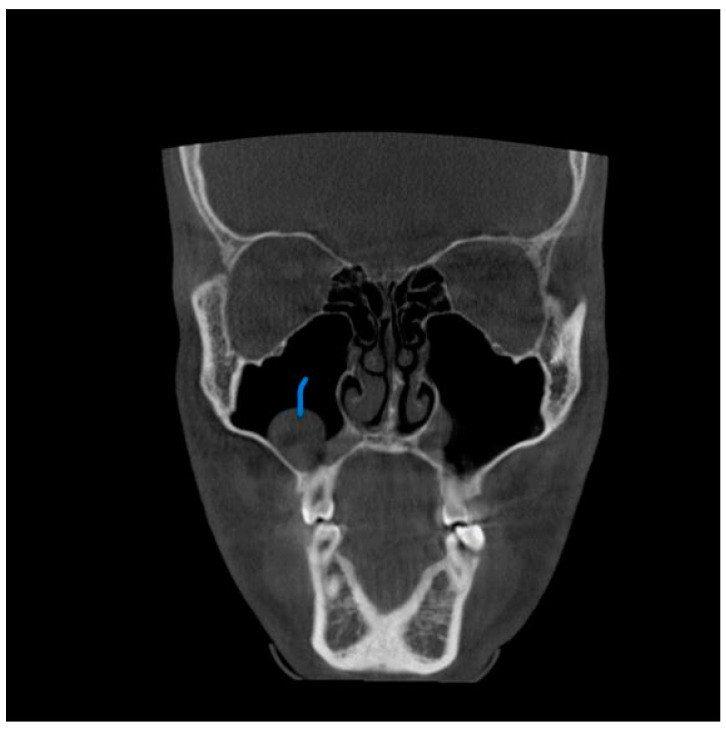
View from the coronal plane. The Blue arrow shows a polypoid growth along the floor of the right maxillary sinuses.

**Figure 4 diagnostics-13-02918-f004:**
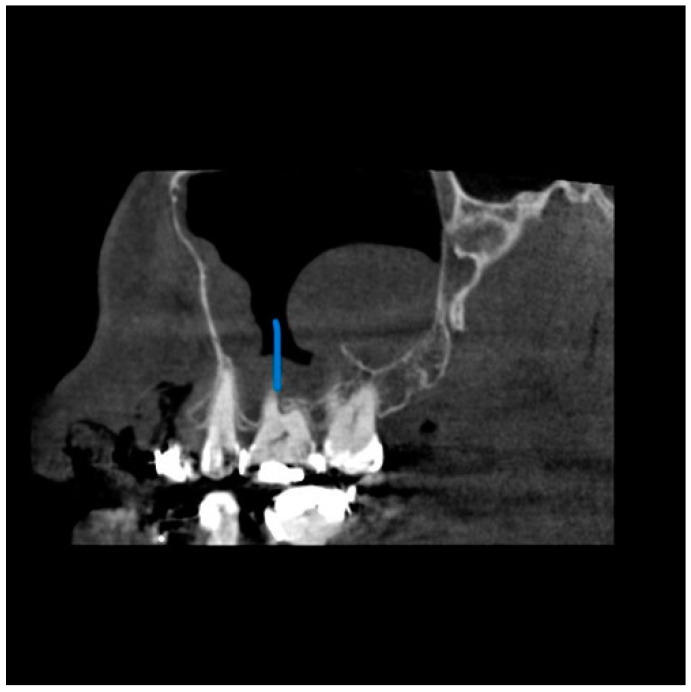
View from the sagittal plane. The Blue arrow shows the close proximity of the periapical lesion in the maxillary molar and the associated sinus wall thickening.

**Figure 5 diagnostics-13-02918-f005:**
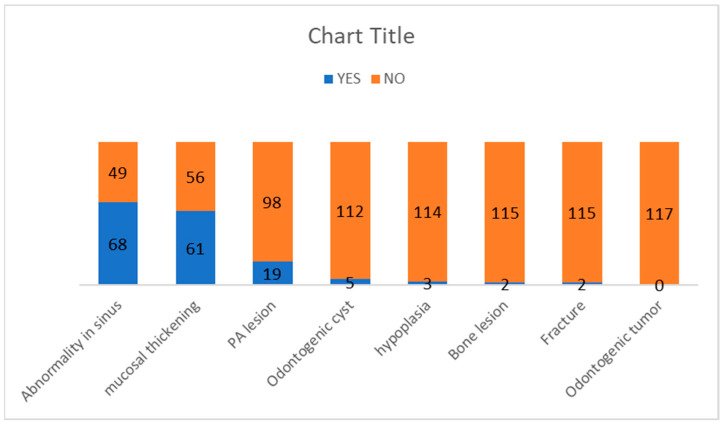
The frequency of the pathologies seen as incidental findings in the maxillary sinus.

**Table 1 diagnostics-13-02918-t001:** Sex-wise distribution of pathologies.

		SEX		Chi-Square	*p*-Value
		F	M		
		Count	Count		
Abnormality in sinus	NO	27	22	0.1510	0.6980
	YES	35	33		
Mucosal thickening	NO	31	25	0.2410	0.6230
	YES	31	30		
Odontogenic cyst	NO	59	53	0.1030	0.7480
	YES	3	2		
Odontogenic tumor	NO	62	55	0	0
	YES	0	0		
Bone lesion	NO	61	54	0.0070	0.9320
	YES	1	1		
Fracture	NO	61	54	0.0070	0.9320
	YES	1	1		
Hypoplasia	NO	60	54	0.2310	0.6310
	YES	2	1		
Periapical lesion	NO	50	48	0.9410	0.3320
	YES	12	7		
Periapical grade	1	5	5	2.4040	0.3010
	2	5	1		
	3	1	0		

**Table 2 diagnostics-13-02918-t002:** Age-wise distribution of pathologies.

		AGE												Chi-Square	*p*-Value
		≤20		21–30		31–40		41–50		51–60		>61			
		Count	Column N %	Count	Column N %	Count	Column N %	Count	Column N %	Count	Column N %	Count	Column N %		
Abnormality in sinus	NO	15	46.9%	7	30.4%	6	54.5%	4	26.7%	14	70.0%	3	18.8%	13.7310	0.017
	YES	17	53.1%	16	69.6%	5	45.5%	11	73.3%	6	30.0%	13	81.3%		
Mucosal thickening	NO	16	50.0%	10	43.5%	6	54.5%	6	40.0%	14	70.0%	4	25.0%	8.0830	0.1520
	YES	16	50.0%	13	56.5%	5	45.5%	9	60.0%	6	30.0%	12	75.0%		
Odontogenic cyst	NO	31	96.9%	21	91.3%	11	100.0%	14	93.3%	20	100.0%	15	93.8%	2.9490	0.7080
	YES	1	3.1%	2	8.7%	0	0.0%	1	6.7%	0	0.0%	1	6.3%		
Odontogenic tumor	NO	32	100.0%	23	100.0%	11	100.0%	15	100.0%	20	100.0%	16	100.0%	00	0.00
	YES	0	0.0%	0	0.0%	0	0.0%	0	0.0%	0	0.0%	0	0.0%		
Bone lesion	NO	32	100.0%	22	95.7%	11	100.0%	14	93.3%	20	100.0%	16	100.0%	4.5210	0.4770
	YES	0	0.0%	1	4.3%	0	0.0%	1	6.7%	0	0.0%	0	0.0%		
Fracture	NO	32	100.0%	22	95.7%	11	100.0%	14	93.3%	20	100.0%	16	100.0%	4.5210	0.4770
	YES	0	0.0%	1	4.3%	0	0.0%	1	6.7%	0	0.0%	0	0.0%		
Opacification	NO	31	96.9%	23	100.0%	11	100.0%	15	100.0%	19	95.0%	15	93.8%	2.6750	0.7500
	YES	1	3.1%	0	0.0%	0	0.0%	0	0.0%	1	5.0%	1	6.3%		
Periapical lesion	NO	26	81.3%	19	82.6%	10	90.9%	11	73.3%	18	90.0%	14	87.5%	2.5200	0.7730
	YES	6	18.8%	4	17.4%	1	9.1%	4	26.7%	2	10.0%	2	12.5%		
Periapical grade	1	2	50.0%	2	50.0%	0	0.0%	4	100.0%	0	0.0%	2	100.0%	12.8920	0.2300
	2	2	50.0%	1	25.0%	1	100.0%	0	0.0%	2	100.0%	0	0.0%		
	3	0	0.0%	1	25.0%	0	0.0%	0	0.0%	0	0.0%	0	0.0%		

## Data Availability

Data supporting reported results can be found, in preprints of this journal including the archived datasets analyzed or generated during the study.

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
