# Peer review of "Prevalence of Incidental Maxillary Sinus Anomalies on CBCT Scans: A Radiographic Study"

_diagnostics, 2023, doi:10.3390/diagnostics13182918_

Round 1

Reviewer 1 Report

This study, entitled “Title- Prevalence of Maxillary Sinus Anomalies on CBCT Scans: A Radiographic Study”, investigated the frequency, type, and location of maxillary sinus abnormalities on CBCT images, and correlated these findings with periapical lesions in the posterior maxilla. Some issues should be addressed as listed below. 

Title: “Title-” is redundant.

Introduction

1.     Page 1 line 38: the full term of PA should be given at its first mention.

2.     Page 1 line 42: Please keep the citation fashion consistent through the entire manuscript, i.e., the full stop is placed before or after the reference.

3.     Pages 1-2 lines 42-44: CBCT scans for dental implant assessment are not recommended to cover the entire maxillary sinus, resulting in partial coverage of the sinus. Please modify the description. Reference: Jacobs R, Salmon B, Codari M, Hassan B, Bornstein MM. Cone beam computed tomography in implant dentistry: recommendations for clinical use. BMC Oral Health. 2018 May 15;18(1):88. doi: 10.1186/s12903-018-0523-5. PMID: 29764458; PMCID: PMC5952365.

4.     Page 2 line 44: the square brackets for reference 3 are missing. 

M&M

1.     Page 2 lines 66-67: “A statistician was consulted and considering the power of the study to be 80%, it was advised to include 117 patient scans.” Please specify which formula and clinical variable(s) (e.g., incidence of specific sinus pathology or periapical lesion?) were used for sample size calculation. Current description regarding the sample size calculation is not acceptable.

2.     Page 2 line 80: please describe the qualification of the two examiners. 

3.     Page 2 lines 87-88: how did the examiners determine if the dimension of the sinus is increased or decreased? How did the examiners determine if the density of the cortical bone of the sinus is changed? 

4.     Page 3 line 98: how did the author consider a missing tooth in the posterior maxilla as the tooth may be removed due to a periapical lesion that could have an impact on the sinus health/pathology.

5.     Page 3 line 109: “Pearson's correlation coefficient was applied to assess the correlation between dimensions” What dimensions? were there any continuous variables in this study? The authors should clarify the nature of all variables assessed. Which variable is continuous, binary, or ordinal?

Results

1.     Were all statistical analyses performed on the patient level or the sinus level. Please clarify which statistical outputs are obtained on the patient level and which ones on the sinus level.

2.     The Chart title shown within the Graph 1 is missing. It is currently shown as “Chart Title”. 

3.     Figure 4: please indicate the location of the periapical lesion in the image. 

Discussion

1.     Page 7 line 180: the square brackets for reference 18 are missing. 

Author Response

Response to the reviewers

We thank our esteemed Reviewer for the thoughtful insight and valuable suggestions.

We have revised our manuscript accordingly and point-by-point responses are provided below. The text has been added or modified from the original text and is highlighted in the revised manuscript. Upon review of our revised manuscript, we hope that you will find it acceptable for publication in your esteemed journal. We look forward to your response.

Reviewer Comments:

Referee 1:

Title: “Title-” is redundant.

Changed.  

Page 1 line 38: the full term of PA should be given at its first mention.

2.     Page 1 line 42: Please keep the citation fashion consistent through the entire manuscript, i.e., the full stop is placed before or after the reference.

3.     Pages 1-2 lines 42-44: CBCT scans for dental implant assessment are not recommended to cover the entire maxillary sinus, resulting in partial coverage of the sinus. Please modify the description. Reference: Jacobs R, Salmon B, Codari M, Hassan B, Bornstein MM. Cone beam computed tomography in implant dentistry: recommendations for clinical use. BMC Oral Health. 2018 May 15;18(1):88. doi: 10.1186/s12903-018-0523-5. PMID: 29764458; PMCID: PMC5952365.

4.     Page 2 line 44: the square brackets for reference 3 are missing.

Changes have been incorporated

Page 2 line 80: please describe the qualification of the two examiners. Age group is unclear, e.g. 2) 20-30; 3)30-40. How about a patient aged 30 years old exactly?

Changes have been incorporated

Results: Tables 1&2: Please spell out PA.

All Graphs should be called Figures.

The data from so-called Graph 1 should be presented into Table 1.

Changes have been incorporated

"A statistician was consulted and considering the power of the study to be 80%, it was advised to include 117 patient scans". Please report the details of the sample size calculation.

Using the formula   with the mini-mum percentage difference to be deemed clinically significant as 8.1%, the sample size required would be 117 for the study.

Page 2 lines 87-88: how did the examiners determine if the dimension of the sinus is increased or decreased? How did the examiners determine if the density of the cortical bone of the sinus is changed?

4.     Page 3 line 98: how did the author consider a missing tooth in the posterior maxilla as the tooth may be removed due to a periapical lesion that could have an impact on the sinus health/pathology.

5.     Page 3 line 109: “Pearson's correlation coefficient was applied to assess the correlation between dimensions” What dimensions? were there any continuous variables in this study? The authors should clarify the nature of all variables assessed. Which variable is continuous, binary, or ordinal?

Linear and angular measurements were done using romexis software of the CBCT Machine. Density was assessed using Hounsefield Units

Edentulous areas and edentulous patients were excluded from the study.

“Pearson's correlation coefficient was applied to assess the correlation between dimensions” What dimensions?

These statements have been deleted and this is part of the second study

Were all statistical analyses performed on the patient level or the sinus level. Please clarify which statistical outputs are obtained on the patient level and which ones on the sinus level.

The Chart title shown within the Graph 1 is missing. It is currently shown as “Chart Title.

Figure 4: please indicate the location of the periapical lesion in the image.

statistical analyses performed on the sinus level.

Changes done

Page 7 line 180: the square brackets for reference 18 are missing.

Done

Reviewer 2 Report

This manuscript was easy to read. Please revise it according to the following comments.

The affiliation numbering can be clarified. There is basically one affiliation only. If all 1-5 refers to the same affiliation, number 1 is enough. There is a "2" at "Manipal Karnataka, India 2", please see if this "2" should be removed.

Introduction: "In another study, the prevalence of flat mucosal thickening ranged from..." Please provide a reference.

"The inability of oral radiologists to detect incidental abnormalities in the maxillary sinus despite being visible in the CBCT volume..." The inability should apply to the general dentists, not oral and maxillofacial radiologists.

Methods: "A statistician was consulted and considering the power of the study to be 80%, it was advised to include 117 patient scans". Please report the details of the sample size calculation.

Max sinus should be maxillary sinus.

"a CBCT unit (Planmeca, Helsinki, Finland)". Please report the model name as well.

Age group is unclear, e.g. 2) 20-30; 3)30-40. How about a patient aged 30 years old exactly?

Results: Tables 1&2: Please spell out PA.

All Graphs should be called Figures.

The data from so-called Graph 1 should be presented into Table 1.

"Patients were found to have pathologic findings in one or both sinuses, such as mucosal thickening, retention cysts, opacification, sinus polyps, and antrolith. (58.1%) (Figures 1,2,3,4)". This line mentioned 5 conditions, not all of which were illustrated by Figures 1-4. Please make the line and the figures consistent to each other. 

Discussion: "CBCT, a new 3D imaging modality". First CBCT introduced by NewTom was available in 1998. Delete "new".

"The study assessed the prevalence of incidental findings in the Maxillary Sinus using cone-beam computerized tomography scans of patients referred to the department." Maxillary Sinus should be in lowercase. "cone-beam computerized tomography" should be CBCT.

Author Response

Response to the reviewers

We thank our esteemed Reviewer for the thoughtful insight and valuable suggestions.

We have revised our manuscript accordingly and point-by-point responses are provided below. The text has been added or modified from the original text and is highlighted in the revised manuscript. Upon review of our revised manuscript, we hope that you will find it acceptable for publication in your esteemed journal. We look forward to your response.

Reviewer Comments:

Referee 2:

This manuscript was easy to read. Please revise it according to the following comments.

The affiliation numbering can be clarified. There is basically one affiliation only. If all 1-5 refers to the same affiliation, number 1 is enough. There is a "2" at "Manipal Karnataka, India 2", please see if this "2" should be removed.

Thank you for the compliment!

The affiliation has been modified

Introduction: "In another study, the prevalence of flat mucosal thickening ranged from..." Please provide a reference.

"The inability of oral radiologists to detect incidental abnormalities in the maxillary sinus despite being visible in the CBCT volume..." The inability should apply to the general dentists, not oral and maxillofacial radiologists.

Changes have been incorporated

Methods:

Max sinus should be maxillary sinus.

"a CBCT unit (Planmeca, Helsinki, Finland)". Please report the model name as well.

Age group is unclear, e.g. 2) 20-30; 3)30-40. How about a patient aged 30 years old exactly?

Changes have been incorporated

Results: Tables 1&2: Please spell out PA.

All Graphs should be called Figures.

The data from so-called Graph 1 should be presented into Table 1.

Changes have been incorporated

"A statistician was consulted and considering the power of the study to be 80%, it was advised to include 117 patient scans". Please report the details of the sample size calculation.

Using the formula  with the mini-mum percentage difference deemed clinically significant as 8.1%, the sample size required would be 117 for the study. it was advised to include 117 patient scans.

 "Patients were found to have pathologic findings in one or both sinuses, such as mucosal thickening, retention cysts, opacification, sinus polyps, and antrolith. (58.1%) (Figures 1,2,3,4)". This line mentioned 5 conditions, not all of which were illustrated by Figures 1-4. Please make the line and the figures consistent to each other.

Mentioned and highlighted

Discussion: "CBCT, a new 3D imaging modality". First CBCT introduced by NewTom was available in 1998. Delete "new".

"The study assessed the prevalence of incidental findings in the Maxillary Sinus using cone-beam computerized tomography scans of patients referred to the department." Maxillary Sinus should be in lowercase. "cone-beam computerized tomography" should be CBCT.

Deleted New

Changes done

Reviewer 3 Report

Dear authors, many thanks for submitting such type of work. The title and abstract are consistent. The study design is adequate. The methodology and the results reported are solid. 

The discussion could be improved. 

Author Response

Response to the reviewers

We thank our esteemed Reviewer for the thoughtful insight and valuable suggestions.

We have revised our manuscript accordingly and point-by-point responses are provided below. The text has been added or modified from the original text and is highlighted in the revised manuscript. Upon review of our revised manuscript, we hope that you will find it acceptable for publication in your esteemed journal. We look forward to your response.

Reviewer Comments:

Referee 3

Dear authors, many thanks for submitting such type of work. The title and abstract are consistent. The study design is adequate. The methodology and the results reported are solid.

The discussion could be improved. – The discussion has been improved with the addition of 3 references. Thank you for the support.